# Hal-Eval: A Universal and Fine-grained Hallucination Evaluation Framework for Large Vision Language Models

Submission Id: 1093

## ABSTRACT

Large Vision-Language Models (LVLMs) exhibit remarkable capabilities but struggle with "hallucinations"—inconsistencies between images and their descriptions. Previous hallucination evaluation studies on LVLMs have identified hallucinations in terms of objects, attributes, and relations but overlooked complex hallucinations that create an entire narrative around a fictional entity. In this paper, we introduce a refined taxonomy of hallucinations, featuring a new category: **Event Hallucination**. We then utilize advanced LLMs to generate and filter fine-grained hallucinatory data consisting of various types of hallucinations, with a particular focus on event hallucinations, laying the groundwork for integrating discriminative and generative evaluation methods within our universal evaluation framework. The proposed benchmark distinctively assesses LVLMs' ability to tackle a broad spectrum of hallucinations, making it a reliable and comprehensive tool for gauging LVLMs' efficacy in handling hallucinations. We will release our code and data.

## KEYWORDS

Hallucination, Event, Evaluation, Large Vision Language Models.

## 1 INTRODUCTION

Large Language Models (LLMs) such as GPT-4 [30], LLaMA [40], and LLaMA2 [41] have markedly enhanced capabilities in natural language understanding (NLU) and generation (NLG). Building on these advancements, recent Large Vision-Language Models (LVLMs) have shown increased proficiency in handling both textual and visual information, sparking significant interest among researchers. [2, 6, 9, 18, 24, 26, 45, 47].

Despite the promising developments in LVLMs, they broadly face the pivotal obstacle of hallucination, which refers to the discrepancy between factual content in images and the associated generated textual descriptions. As hallucination poses significant concerns for the LVLMs' reliability and robustness [12, 20, 23, 28, 39, 42, 46], researchers have devised strategies for hallucinations evaluation to bolster the practical deployment of LVLMs, including discriminative and generative methods. The former method directly prompts candidate LVLMs to determine the presence of a particular hallucination, whereas the latter assesses the text produced by these candidate LVLMs.

*ACM MM, 2024, Melbourne, Australia*

© 2024 Copyright held by the owner/author(s). Publication rights licensed to ACM.

ACM ISBN 978-x-xxxx-xxxx-x/YY/MM

https://doi.org/10.1145/nnnnnnn.nnnnnnn

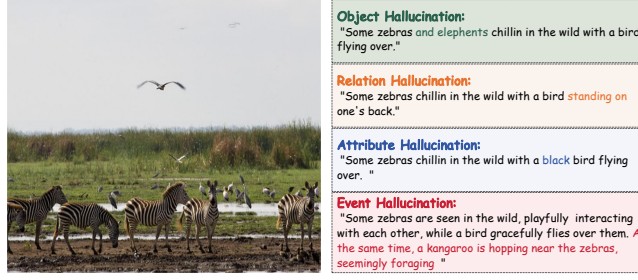

**Figure 1: Different types of hallucination. Event hallucination, which involves more complex vision-language discrepancy compared to other types of hallucination, is commonly overlooked by previous efforts.**

Prior research [13, 17, 28] has delineated vision-language mismatches as issues of non-existent objects, incorrect object attributes, or inaccurate object relations, yet it does not encompass the full spectrum of hallucinations observed in LVLMs. For example, as depicted in Figure 1, LVLM outputs can exhibit more intricate hallucinations, such as "At the same time, a kangaroo is hopping near the zebras, seemingly foraging." This type of hallucination invents a fictional target and weaves an entire narrative around it, including its attributes, relationships, and actions. We categorize these intricate narratives as **event hallucinations**. Our preliminary experiments indicate that the occurrence of event hallucinations significantly escalates as the output length of LVLMs increases, underscoring its significance as a hallucination phenomenon that warrants attention. However, there is an absence of fine-grained hallucination evaluation benchmarks for LVLMs that comprehensively address the various types of hallucinations—such as artificial objects, relationships, attributes, and events—while also accommodating both discriminative and generative evaluation methods.

To this end, we propose a universal, fine-grained hallucination evaluation framework for LVLMs. This framework comprehensively evaluates a broad spectrum of hallucination types, encompassing objects, relationships, attributes, and notably, **events** with discriminative and generative evaluation methodologies. Specifically, we first develop an automatic annotation pipeline for fine-grained hallucinations, which leverages the sophisticated capabilities of GPT4 to generate and filter hallucinatory data. This pipeline then serves as a solid foundation for unifying discriminative and generative evaluation methodologies in our framework:

- For the discriminative evaluation, we construct a dataset that features image captions with hallucinations generated through our pipeline. Candidate LVLMs are presented with uniform question templates to determine if a given caption, produced by us, manifests a specific type of hallucination relative to the image content.

| Benchmark | Tasks | | Discriminative Hallucination | | | | Generative Hallucination | | | |
|---|---|---|---|---|---|---|---|---|---|---|
| | Dis | Gen | Object | Attribute | Relation | Event | Object | Attribute | Relation | Event |
| POPE [20] | ✓ | ✗ | ✓ | ✗ | ✗ | ✗ | ✗ | ✗ | ✗ | ✗ |
| NOPE [29] | ✓ | ✗ | ✓ | ✗ | ✗ | ✗ | ✗ | ✗ | ✗ | ✗ |
| CIEM [14] | ✓ | ✗ | ✓ | ✗ | ✗ | ✗ | ✗ | ✗ | ✗ | ✗ |
| M-HalDetect [13] | ✗ | ✓ | ✗ | ✗ | ✗ | ✗ | ✓ | ✓ | ✓ | ✗ |
| GAVIE [23] | ✗ | ✓ | ✗ | ✗ | ✗ | ✗ | ✓ | ✓ | ✗ | ✗ |
| FAITHScore [17] | ✗ | ✓ | ✗ | ✗ | ✗ | ✗ | ✓ | ✓ | ✓ | ✗ |
| HaELM [44] | ✗ | ✓ | ✗ | ✗ | ✗ | - | - | - | - | ✗ |
| MMHal-Bench [38] | ✗ | ✓ | ✗ | ✗ | ✗ | - | - | - | - | ✗ |
| AMBER [43] | ✓ | ✓ | ✓ | ✓ | ✓ | ✗ | ✓ | ✗ | ✗ | ✗ |
| Hal-Eval | ✓ | ✓ | ✓ | ✓ | ✓ | ✓ | ✓ | ✓ | ✓ | ✓ |

**Table 1: Comparison of Hallucination Evaluation Benchmarks for LVLMs.**

- For the generative evaluation, our pipeline facilitates the creation of a large-scale hallucinatory dataset. This dataset serves to fine-tune an LVLM into a specialized evaluator, Hal-Evaluator. This evaluator assesses LVLM-generated descriptions and associated images, identifying various hallucination types without needing reference captions.

We conduct thorough experiments and analysis with six leading LLMs within our framework, assessing their performance in terms of hallucination under both discriminative and generative paradigms. Our key findings are:

- The existing three categories of hallucinations (object, attribute, relation) overlook the existence of event-type hallucinations and are, therefore, insufficient to encompass all types of hallucinations.
- Utilizing Chain-of-Thought (COT) significantly helps models minimize hallucinations during discriminative evaluations, particularly those involving relationships and events.
- The incidence of hallucinations, especially event hallucinations, increases with the length of the output. Length control becomes a crucial aspect of generative evaluations, affecting comparative performance trends among LVLMs under varied output lengths.
- The suitability of evaluation methodology varies according to the type of hallucinations. Using discriminative and generative evaluations together gives a fuller view of tendencies to LVLM hallucination.
- The hallucinatory samples used to train our evaluator also serve as effective supervised fine-tuning data for LVLMs, contributing to reducing hallucinations and enhancing their benchmark performance.

In summary, we introduce a novel hallucination category (event hallucination) of LVLMs, a universal and fine-grained evaluation framework for LVLMs that spans various hallucination types and unifies discriminative and generative approaches (as shown in Table 1), along with some groundbreaking insights to guide future research on vision-language hallucination.

## 2 PRELIMINARY: HALLUCINATION IN LVLMS

Previous works have characterized misalignment of hallucination as claims of non-existent objects, incorrect object attributes, or inaccurate object relations. However, we find that this only partially

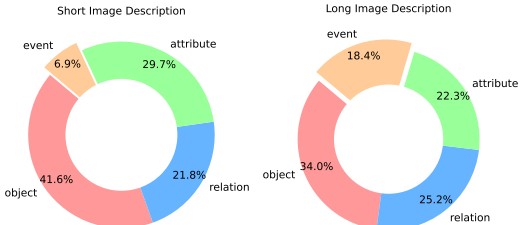

**Figure 2: The left sub-figure shows the ratios of various hallucinations in mPLUG-owl's image descriptions with token lengths under 20. The right sub-figure presents these ratios for descriptions exceeding 20 tokens.**

covers the spectrum of hallucinations present in LVLMs. For instance, as shown in the left part of Figure 8, the outputs of LVLMs include more complex hallucinations: " to chase after a squirrel that has invaded its playtime ." We refer to these complex hallucinations as event hallucinations. To further clarify the concept of different hallucinations, we provide strict definitions for four different types of hallucinations in this paper:

**Object hallucination:** The LVLM inaccurately describes an object that does not exist in the image. This could be a misidentification, where the model correctly detects an object's presence but incorrectly labels it, or an additional non-existent object, where the model asserts the presence of an object that does not exist.

**Attribute hallucination:** The LVLM correctly identifies an object *that exists in the image*, but inaccurately describes the attributes of that object. Attributes could include color, size, shape, position, or any other characteristic that defines the object.

**Relation hallucination:** The LVLM incorrectly describes the relationship between two or more objects *that clearly exists in the image*. This could involve misrepresenting spatial relationships (e.g., describing an object as being on top of another when it's actually beside it), functional relationships (e.g., stating that a person is riding a bicycle when they are standing next to it), or other types of interactions or connections between objects.

**Event hallucination:** The LVLM not only describes a non-existent target but also constructs complete events around the non-existent target, including its attributes, relations, and actions. This type of hallucination involves a complex interplay of objects, attributes, and relations and often forms a narrative or sequence of actions that does not align with the actual content of the image.

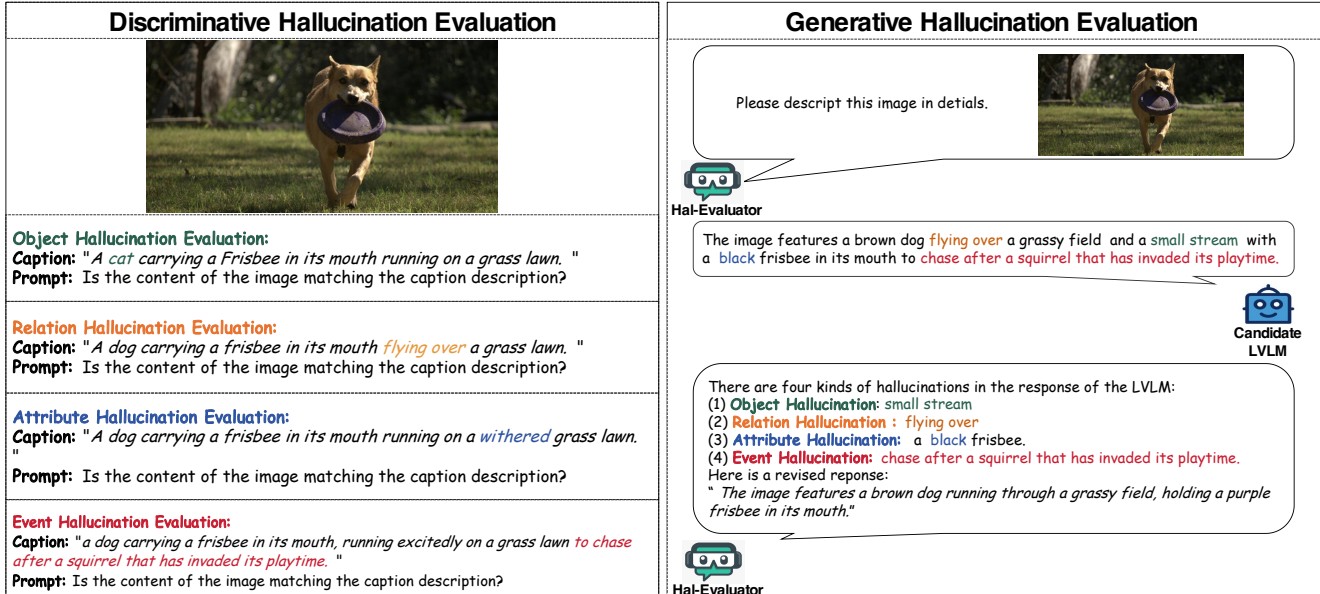

**Figure 3: This figure provides a schematic of the discriminative evaluation and generative evaluation used in Hal-Eval.**

Building upon these definitions, we further investigate the proportion of different types of hallucinations present within the output of LVLMs. As depicted in Figure 2, we collected 5,000 image-caption pairs from COCO [22]. We had them described by mPLUG-owl [45] and LLaVA [26], respectively. Subsequently, we provided both the ground truth image descriptions and the model-generated descriptions to GPT-4 [30], prompting it to inspect whether these descriptions encompassed hallucinations and to categorize them based on Object, Attribute, Relationship, and Event hallucinations (For more details about the experiment, please refer to our Supplemental Material D.1.). We tabulated the proportions of different types of hallucinations at varying description lengths. As Figure 2 demonstrates, we noted a significant increase in the share of event hallucinations as the length of the description extends. This experimental observation substantiates our finding: **The existing three categories of hallucinations (object, attribute, relation) overlook the existence of event-type hallucinations. They are, therefore, insufficient to encompass all types of hallucinations.**

## 3 METHOD

We proposed a comprehensive and universal hallucination evaluation benchmark, Hal-Eval. As shown in Figure 3, Hal-Eval includes both Discriminative Evaluation and Generative Evaluation and can effectively evaluate different types of hallucinations. In the following subsection, we will initially introduce a fine-grained hallucination annotation pipeline. This pipeline is employed to construct both the evaluation dataset for Hal-Eval and a large-scale hallucination detection dataset known as Hal-Data. Hal-Data, in turn, serves as the training data for the Hal-evaluator, an evaluation model designed to perform generative hallucination evaluations. Subsequently, we will delve into an in-depth discussion of the discriminative evaluation process. Finally, we will elucidate the training procedure for the Hal-evaluator and outline how generative evaluations are conducted using this model.

### 3.1 Automatic Fine-grained Hallucination Annotation Pipeline

The existing multimodal hallucination research lacks large-scale datasets with fine-grained annotations specific to hallucinations. To address this issue, we design an automatic fine-grained hallucination annotation pipeline featuring annotations for four hallucination types and specific hallucination content.

**Data Annotation.** We annotated image-text paired data based on GPT-4. We initially established a rigorous definition for various types of hallucinations as we mentioned in Section 2. Building upon this groundwork, we engaged GPT-4 to rephrase the collated image-text pairs in line with the diverse classifications of hallucinations. This step involved injecting distinctive hallucinatory elements into the original captions. The outcome of this procedure was a collection of image descriptions enriched with specified hallucination categories. Moreover, we delegated to GPT-4 the responsibility of annotating the position of specific hallucinatory content in the image description. Please refer to the supplemental material C.2 for more details.

**Data Filtering.** Following the initial annotation phase, we identified that the quality of the labeled data remained unsatisfactory. Random sampling revealed that approximately 30% of the annotated dataset still harbored noise that failed to meet our stringent labeling criteria. Hence, we proceeded to craft a tailored prompt to commission GPT-4 for the task of purging and refining the noisy annotations, a process thoroughly outlined in the supplemental material C.3. Subsequent to GPT-4's meticulous cleanup operation, a manual verification process ascertained that over 97% of the data accorded with the stipulated annotation standards.

### 3.2 Discriminative Evaluation

*3.2.1 Constructing Evaluation Dataset.* **Data Collection:** Previous benchmarks such as POPE [20] predominantly utilized manually annotated datasets like COCO [22]. However, the COCO dataset is frequently employed to construct general benchmarks such as

VQA v2 and visual grounding. These benchmarks are often used for instruction finetuning of LVLM models, which results in evaluation data being in the same domain as the models' finetuning data. This overlap hinders a true assessment of the models' zero-shot hallucination capabilities. To address this issue, we divided our evaluation dataset into two parts: in-domain data from the COCO 2014 validation and COCO 2017 test sets and an out-of-domain dataset randomly sampled from web-based data like CC [5], SBU [32], LAION [36].

**Data Annotation:** We used the automatic annotation process detailed in Section A to annotate both in-domain and out-of-domain evaluation datasets, resulting in 5,000 detailed annotations each. These annotations identify hallucination types and content. An annotated sample is represented as $S = \{I, C^T, C^O, C^R, C^E, C^A\}$, where I is the image, $C^T$ is the correct image caption, and $C^O, C^R, C^E, C^A$ denote captions with Object, Relation, Event, and Attribute hallucinations, respectively.

*3.2.2 Evaluation Process.* In prior work, the discriminative evaluation method proposed for evaluating a specific type of hallucination asked LVLMs if the content of that type existed in the image. For instance, evaluating object hallucinations inquires about the presence of a specific object. In contrast, we have proposed a more natural questioning method, which is as follows:

**Prompting LVLMs.** Assuming a sample as S, the form of the prompt is as follows:

*<Image> I*

*Caption: $C \in \{C^T, C^O, C^R, C^E, C^A\}$.*

*Question: Does the description in the caption accurately reflect the content of the image?*

By controlling the different types of caption $C$, we can evaluate different types of hallucinations based on a unified prompt template. For example, we can set $C = C^A$ to evaluate Attribute-type hallucinations.

**Evaluation Metric.** Similar to POPE [20], we also use Accuracy, Precision, Recall, F1 score, and "Yes" ratio as the evaluation metrics. Here, Accuracy represents the number of correctly answered cases, while Precision and Recall, respectively, indicate the proportion of correctly answered questions with responses "Yes" or "No." The F1 score integrates the outcomes of Precision and Recall, which we select as the primary evaluation metric. The "Yes ratio" serves as a reference for analyzing model behaviors.

## 3.3 Generative Evaluation

*3.3.1 Overview .* Regarding generative evaluation, current evaluation methods either rely on proprietary models that require subscription fees, such as GPT-4, or depend on fine-tuned large language models (LLMs) that necessitate additional ground truth annotations—a process that is prohibitively expensive. This significantly restricts the scalability of evaluating models. In response, we propose Hal-Evaluator, a reference-free, open-source evaluation model designed specifically to detect hallucinatory content. Hal-Evaluator is fine-tuned on LLaVA 1.5 [24], which is also an LVLM; as illustrated in Figure 3, it takes as input the description of an image provided by the LVLMs under evaluation, as well as the corresponding image itself. Following this, it evaluates whether the description contains

hallucinations. If hallucinations are detected, they provide the specific content and categorization of the hallucinations. Ultimately, it can even modify the hallucinated information in the description to output an accurate depiction. In this way, our generative evaluation eliminates the need for additional reference annotation, enabling hallucination evaluation based solely on the content of the image.

To train the Hal-Evaluator, which is capable of effectively identifying different types of hallucinations, a large-scale, fine-grained hallucinatory image-text dataset is necessary as it facilitates the refinement of training for Hal-Evaluator intended to detect and correct hallucinatory content. However, no dataset of this scale with detailed annotations currently exists. Therefore, we initially constructed **Hal-Data**, the first large-scale, fine-grained dataset with hallucination annotations, based on the pipeline mentioned in Subsection A.

*3.3.2 Instruction finetuning of Hal-Evaluator.* This dataset, referred to as Hal-Data, was generated using an automatic hallucination annotation pipeline and comprises 2 million instances. Hal-Data is split into two parts: Hal-Data 130k, which includes 130,000 GPT-4 annotated and curated image-text pairs, each consisting of an image, a valid image caption, and a hallucination description; and Hal-Data 2M, which includes 2 million image-text pairs created by our caption model trained on the 130,000 high-quality captions from Hal-Data 130k. Below, we detail the creation process for Hal-Data.

**Data Collection for Hal-Data 130k:** To ensure diversity and comprehensiveness, we initially compiled about 200,000 images from various sources, including 80,000 in-domain COCO dataset images [22], 80,000 web images from sources like CC [5], SBU [32], and LAION [36], and 40,000 image-text datasets from ShareGPT4-V [7] to match the style of LVLM outputs. We then used AFHA to annotate this data, resulting in a final collection of 130,000 meticulously annotated GPT4 instances, named Hal-Data 130k.

**Generation for Hal-Data 2M:** We further selected a subset of 2 million image-caption pairs from current public datasets (see Appendix B.1 for more details) and constructed a large-scale hallucination dataset named Hal-Data 2M. Due to the high cost of using GPT-4, we fine-tuned the open-source large-scale language model LLaMA2 13B [40] on Hal-Data 130k and employed it to modify the image captions of Hal-Data 2M by introducing different types of hallucinations and annotating them.

Based on Hal-Data, we fine-tuned LLaVA 1.5 13B [24], recent SOTA LVLM, with 2M instruction data specifically designed for detecting and correcting hallucinations in image captions, leading to the development of Hal-Evaluator. (For more details, please refer to our supplemental material D.1.)

*3.3.3 Generative Evaluation Based in Hal-Evaluator.* As illustrated in Figure 3, the input to Hal-Evaluator consists of two parts: an image and the corresponding textual description by the candidate LVLM to be evaluated. We prompt Hal-Evaluator to first determine if the text description contains hallucinations based on the image content. If hallucinations are detected, Hal-Evaluator will identify the type of hallucination and its content. Ultimately, Hal-Evaluator can also correct the hallucinatory content in the image description, providing a revised depiction of the image.

*3.3.4 Evaluation Metric.* To evaluate the generative hallucination of LVLMs, we prompt them to describe images from both an in-domain 5K dataset and an out-of-domain 5K dataset mentioned in Subsection 3.2.1 with short length and longer length. These descriptions, coupled with the respective images, are then fed into the pre-trained Hal-Evaluator. Our procedure involves prompting Hal-Evaluator to evaluate the existence and category of any hallucinatory content. Accuracy serves as the principal metric for our evaluation, which measures the proportion of outputs correctly identified as free from hallucinations. Suppose the number of all outputs is $N$, the outputs that contain hallucinations are $N_h$, and the accuracy is calculated as $A = \frac{N-N_h}{N}$. Moreover, we track the probability of various types of hallucinations encapsulated in the hallucination ratio. For instance, the number of outputs containing the object hallucination is $N_h^o$, and the object ratio $r_o$ is calculated as $r_O = \frac{N_h^o}{N_h}$.

# 4 EXPERIMENTS

Hal-Eval is divided into two distinct segments: Discriminative Evaluation and Generative Evaluation. We have opted to assess five widely utilized open-source LVLMs: MiniGPT-4 [47], InstructBLIP [9], mPLUG-owl [45], LLaVA [26], LLaVA1.5 [24] and one close-source LVLM: GPT4-V [31].

## 4.1 Discriminative Evaluation

*4.1.1 Main Results.* As shown in Table 2, we evaluate the performance of five models on different types of hallucinations following the method outlined in Subsection 3. First, LLaVA1.5 and LLaVA exhibit a more pronounced predilection for hallucinations when tested against out-of-domain datasets as opposed to in-domain datasets. This trend could possibly be ascribed to the prevalent incorporation of COCO [22] during the instruction tuning phase of the models. Moreover, we noticed that the results derived from the POPE [20] metric indicate a significant tendency among most models to favor "yes" responses. In contrast, within our discriminative evaluations, such a penchant is exclusively noted in the InstructBLIP. This distinction serves to underscore that Hal-eval can effectively avoid the bias of the model towards answering "yes". In the end, our findings indicate that, with the exception of GPT-4, the performance of the currently accessible open-source LVLMs in discriminative evaluations is subpar. These models face challenges in accurately discerning and interpreting hallucinatory content within image descriptions.

*4.1.2 Analysis of Discriminative Evaluation.* **Data Reliability Analysis:** Our proposed evaluation dataset comprises 5,000 in-domain and 5,000 out-of-domain images, which we annotated based on the AFHA framework. To verify the accuracy of the annotations, we randomly sampled 100 cases each from both the in-domain and out-of-domain data for manual validation (Please refer to supplemental material C for more details). We found that after GPT-4's annotation and filtering process, the annotation accuracy rate in the COCO dataset reached 98%. Meanwhile, the annotation accuracy for the out-of-domain dataset stood at 97%. This high level of accuracy in both datasets underscores the effectiveness of our annotation process.

**Effectiveness of Chain-Of-Thought (COT) for Mitigating Discriminative Hallucination:** For discriminative evaluation, we employed a

| Dataset | Type | Model | Accuracy | Precision | Recall | F1 | Yes (%) |
|---|---|---|---|---|---|---|---|
| In-domain | Object | mPLUG-Owl | 49.8 | 49.8 | 44.7 | 47.1 | 44.1 |
| | | LLaVA | 52.6 | 55.5 | 26.3 | 35.7 | 23.6 |
| | | MiniGPT-4 | 50.4 | 50.3 | 46.5 | 48.3 | 40.2 |
| | | InstructBLIP | 50.0 | 50.0 | 99.0 | 66.5 | 98.0 |
| | | LLaVA 1.5 | 62.2 | 76.1 | 35.6 | 48.5 | 23.3 |
| | | GPT4-V | 85.3 | 87.0 | 80.2 | 85.3 | 52.4 |
| | Attribute | mPLUG-Owl | 49.9 | 49.9 | 44.7 | 47.2 | 44.6 |
| | | LLaVA | 52.8 | 55.9 | 26.3 | 35.8 | 23.5 |
| | | MiniGPT-4 | 51.1 | 51.1 | 46.5 | 48.7 | 39.4 |
| | | InstructBLIP | 49.8 | 49.8 | 99.0 | 66.3 | 98.1 |
| | | LLaVA 1.5 | 62.2 | 76.1 | 35.6 | 48.5 | 23.3 |
| | | GPT4-V | 84.1 | 88.2 | 79.3 | 83.7 | 48.3 |
| | Relation | mPLUG-Owl | 50.4 | 50.5 | 44.7 | 47.4 | 44.7 |
| | | LLaVA | 52.7 | 55.7 | 26.3 | 35.8 | 23.7 |
| | | MiniGPT-4 | 50.4 | 50.1 | 46.5 | 48.2 | 40.0 |
| | | InstructBLIP | 49.8 | 49.9 | 99.0 | 66.3 | 97.7 |
| | | LLaVA 1.5 | 55.4 | 59.1 | 35.6 | 44.4 | 22.1 |
| | | GPT4-V | 83.5 | 80.2 | 88.7 | 83.3 | 49.2 |
| | Event | mPLUG-Owl | 49.7 | 49.7 | 44.6 | 47.0 | 44.8 |
| | | LLaVA | 51.5 | 53.0 | 26.3 | 35.1 | 24.8 |
| | | MiniGPT-4 | 32.6 | 50.0 | 46.5 | 48.2 | 40.3 |
| | | InstructBLIP | 49.6 | 49.7 | 99.0 | 66.2 | 84.3 |
| | | LLaVA 1.5 | 62.7 | 77.9 | 45.6 | 58.9 | 22.8 |
| | | GPT4-V | 86.3 | 86.1 | 80.5 | 87.2 | 51.6 |
| Out-of-domain | Object | mPLUG-Owl | 50.3 | 50.4 | 43.6 | 46.8 | 43.4 |
| | | LLaVA | 50.7 | 52.7 | 9.0 | 15.3 | 7.2 |
| | | MiniGPT-4 | 50.3 | 51.7 | 53.6 | 52.6 | 25.0 |
| | | InstructBLIP | 50.0 | 50.0 | 100.0 | 66.6 | 100.0 |
| | | LLaVA 1.5 | 59.2 | 86.2 | 21.9 | 35.0 | 18.2 |
| | | GPT4-V | 84.7 | 87.4 | 80.9 | 86.1 | 51.4 |
| | Attribute | mPLUG-Owl | 50.4 | 50.5 | 43.6 | 46.8 | 42.9 |
| | | LLaVA | 51.8 | 66.5 | 9.0 | 15.8 | 6.2 |
| | | MiniGPT-4 | 50.0 | 51.5 | 53.6 | 52.6 | 24.7 |
| | | InstructBLIP | 50.0 | 50.0 | 100.0 | 66.6 | 100.0 |
| | | LLaVA 1.5 | 58.1 | 79.4 | 21.9 | 34.4 | 13.8 |
| | | GPT4-V | 82.6 | 80.1 | 79.5 | 81.5 | 48.3 |
| | Relation | mPLUG-Owl | 50.0 | 50.0 | 43.6 | 46.6 | 43.1 |
| | | LLaVA | 50.8 | 57.1 | 9.0 | 15.5 | 7.8 |
| | | MiniGPT-4 | 49.7 | 50.9 | 53.6 | 52.2 | 24.6 |
| | | InstructBLIP | 50.0 | 50.0 | 100 | 66.6 | 100.0 |
| | | LLaVA 1.5 | 53.7 | 60.2 | 21.9 | 32.2 | 12.7 |
| | | GPT4-V | 84.0 | 81.1 | 87.5 | 83.5 | 50.3 |
| | Event | mPLUG-Owl | 50.1 | 50.1 | 43.6 | 46.6 | 43.3 |
| | | LLaVA | 46.2 | 31.2 | 9.0 | 14.0 | 13.2 |
| | | MiniGPT-4 | 49.3 | 52.3 | 53.6 | 53.0 | 24.3 |
| | | InstructBLIP | 50.0 | 50.0 | 100 | 66.6 | 99.9 |
| | | LLaVA 1.5 | 57.7 | 77.2 | 41.9 | 44.2 | 14.2 |
| | | GPT4-V | 85.3 | 83.2 | 84.5 | 84.5 | 50.3 |

**Table 2: Results of LVLMs under evaluation of four hallucination types on the in-domain dataset and out-of-domain dataset. Yes denotes the proportion of answering "Yes" to the given question.**

chain of thought (COT) approach to systematically evaluate whether the LVLM matches the content of images with their respective captions (Refer to Appendix C.4 for more details). As shown in Figure 4, we observed a significant reduction in discriminative hallucinations on both in-domain and out-of-domain datasets after employing COT

| Model | Length | In-domain | | | | | Out-of-domain | | | | |
|---|---|---|---|---|---|---|---|---|---|---|---|
| | | Object Ratio | Relation Ratio | Attribute Ratio | Event Ratio | Acc | Object Ratio | Relation Ratio | Attribute Ratio | Event Ratio | Acc |
| MiniGPT-4 | 28.7 | 36.6 | 30.6 | 16.5 | 10.6 | 69.3 | 45.5 | 20.8 | 19.2 | 14.6 | 66.5 |
| | 79.6 | 46.2 | 22.5 | 8.0 | 23.4 | 61.4 | 53.7 | 9.7 | 7.2 | 29.6 | 50.1 |
| InstructBLIP | 10.3 | 34.2 | 45.2 | 10.3 | 8.3 | 89.1 | 47.6 | 27.4 | 13.2 | 10.2 | 91.0 |
| | 80.6 | 25.7 | 12.6 | 16.8 | 51.3 | 35.5 | 19.6 | 11.4 | 15.2 | 59.3 | 41.3 |
| mPLUG-owl | 28.3 | 45.5 | 24.6 | 16.3 | 13.4 | 45.4 | 40.5 | 21.2 | 18.5 | 19.4 | 43.5 |
| | 78.3 | 46.2 | 9.5 | 12.5 | 31.7 | 27.3 | 45.9 | 9.3 | 4.6 | 40.2 | 29.5 |
| LLaVA | 37.3 | 40.1 | 18.5 | 4.5 | 37.1 | 47.4 | 34.9 | 23.2 | 24.4 | 17.8 | 46.3 |
| | 88.3 | 45.7 | 9.4 | 3.1 | 42.1 | 23.3 | 38.3 | 7.2 | 2.2 | 52.6 | 26.3 |
| LLaVA1.5 | 10.3 | 23.7 | 58.8 | 10.6 | 7.0 | 55.7 | 30.0 | 48.4 | 11.6 | 10.2 | 49.5 |
| | 84.5 | 42.2 | 13.0 | 3.6 | 41.4 | 44.6 | 34.6 | 8.8 | 2.7 | 54.3 | 46.4 |
| GPT4-V | 21.5 | 27.7 | 18.8 | 20.6 | 14.0 | 92.7 | 23.7 | 27.8 | 17.6 | 29.4 | 89.7 |
| | 80.2 | 32.9 | 21.0 | 16.6 | 30.4 | 77.6 | 30.9 | 18.0 | 13.6 | 38.4 | 73.1 |

**Table 3: Generative Hallucination Evaluation for LVLMs.**

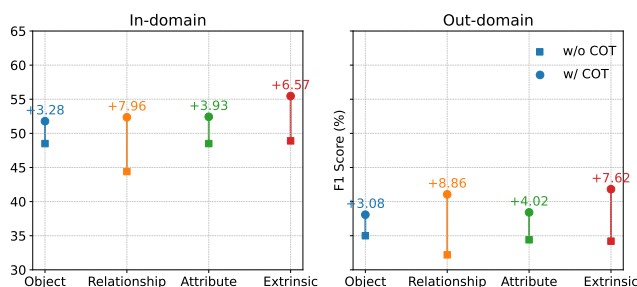

**Figure 4: Comparison of LLaVA1.5 and LLaVA 1.5-COT. We report the F1 score for both of them.**

to LLaVA 1.5. Based on the above experiments, we have the following finding:

> **Finding 1**: *Utilizing COT is particularly effective in reducing discriminative hallucinations for LVLMs, especially for those related to relationships and **events***.

We suggest that the increased effectiveness of COT with relationship and event type hallucinations is due to their intrinsic reliance on contextual understanding and inference-making.

## 4.2 Generative Evaluation

*4.2.1 Main Results.* As indicated in Table 3, our investigation has revealed that contemporary models continue to have a significant inclination toward producing hallucinations. MiniGPT-4 and Instruct-BLIP, displayed robust in-domain accuracy, with the latter achieving 89.1% accuracy when the average output length was approximately 10 tokens. Both mPLUG-owl and LLaVA showed moderate performance across all evaluated metrics, whether tested on in-domain or out-of-domain data. GPT-4V achieved the best result, but we also observed a notable decline in accuracy with increasing output length, accompanied by a significant rise in the proportion of event-type hallucinations. Furthermore, we found that when generating long responses, all models became more prone to producing hallucination content, with the incidence of event hallucinations rising markedly.

*4.2.2 Analysis of Generative Evaluation.* **Correlations with Human Evaluations:** To verify the correlation between the generative

| Metric | type | $r$ (%) | $\rho$ (%) | $\tau$ (%) |
|---|---|---|---|---|
| BLEU-4 | Gen | -1.3 | -7.1 | -4.8 |
| ROUGE-L | Gen | -6.7 | -8.5 | -7.4 |
| GPT4-V | Gen | 42.2 | 38.5 | 31.3 |
| CHAIR | Gen | 17.8 | 19.2 | 18.8 |
| Hal-EML | Gen | 29.8 | 21.6 | 33.7 |
| Hal-Eval-Gen | Gen | **47.34** | **37.20** | **43.43** |

**Table 4: Correlation between each evaluation metric and human judgment on LVLM hallucinations, measured by Pearson's $r$, Spearman's $\rho$, and Kendall's $\tau$.**

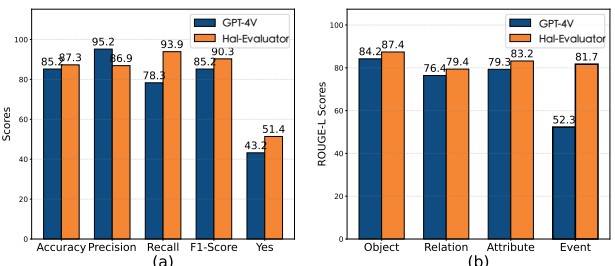

**Figure 5: The left sub-figure displays the results of the discriminative evaluation for GPT-4V and Hal-Evaluator. The right sub-figure compares the ROUGE-L between hallucination content detected by GPT-4V and Hal-Evaluator with the annotated hallucination content.**

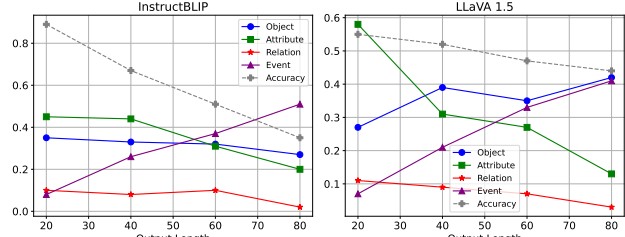

**Figure 6: The figure depicts the proportions of different types of hallucinations in the outputs of InstructBLIP and LLaMA1.5, as well as the gray line illustrating the variation in accuracy.**

evaluation and human judgment, we conduct the following experiments. We first select the test dataset from COCO 2014 [22] for human evaluation. This test set comprises 50 images. Each image is

| Discriminative (Ave F1) | | | | Generative (Hallucination Ratio) | | | |
|---|---|---|---|---|---|---|---|
| Object | Relation | Attribute | Event | Object | Relation | Attribute | Event |
| 52.8 | 52.6 | 51.8 | 54.3 | 38.3 | 12.7 | 8.8 | 41.2 |

**Table 5: This table displays the average F1 scores for various hallucination types in discriminative assessments and the average hallucination rates for LVLM's long outputs (>=80) in generative assessments.**

| Model | Dis | | Gen | | |
|---|---|---|---|---|---|
| | ID | OOD | ID | OOD | length |
| LLaVA1.5 | 47.6 | 34.0 | 47.4 | 46.3 | 10.3 |
| Hal-VL | 69.7 | 71.85 | 70.9 | 60.4 | 10.5 |

**Table 6: The evaluation results of Hal-VL on Hal-Eval. For the discriminative evaluation, we have only listed the average F1 scores for different types of hallucination. For the generative evaluation, we list the accuracy with short output length.**

supplemented with five reference captions and object labels provided by the COCO dataset. We selected three LVLMs – LLaVA [26], mPLUG-owl [45], and instructBLIP [9] – to describe the content of the test set, and we sought the annotation and evaluation from 15 human annotators to evaluate the presence of hallucinations in these data. We made a comparison among five benchmarks: ROUGE-L [21], BLEU-4 [33], CHAIR [35], Hal-EML [44], GPT4-V along with the module of Hal-Eval – Hal-Eval-Generative. Table 4 delineates the correlation between various evaluation metrics and human judgment regarding LVLM faithfulness, gauged using Pearson's $r$, Spearman's $\rho$, and Kendall's $\tau$. Our generative evaluation distinctly stands out and demonstrates a robust positive correlation, underlining the superior alignment with human perceptions.

**Effectiveness of Hal-Evaluator:** To further verify the effectiveness of Hal-Evaluator for hallucination detection, we evaluated Hal-Evaluator and GPT-4V (as candidate LVLMs instead of evaluator here) based on the discriminative evaluation of Hal-Eval scripted in subsection 3.2.2, evaluating the detection of different types of hallucinations in 5K coco data. The results disclosed that Hal-Evaluator outperforms GPT-4V in hallucination detection ability, as shown in the Sub-Figure 5 (a).

**Analysis of Event Hallucination:** We tasked GPT-4V and Hal-Evaluator with detecting the hallucination content in image descriptions of the evaluation dataset of Hal-Eval. We evaluated the overlap between the hallucination content as identified by GPT-4V and Hal-Evaluator and the annotated hallucination content that exists in the descriptions. The overlap was quantified using the ROUGE-L score, as shown in Sub-figure 5 (b). The experimental results show that both GPT-4V and Hal-Evaluator can accurately identify the majority of hallucinated content in image descriptions for the first three types of hallucinations (object, attribute, relation). However, when it comes to event hallucinations, GPT-4V struggles to pinpoint the hallucinated content accurately, while Hal-Evaluator demonstrates a reliable identifying capability. Considering that GPT-4V is the current top-performing LVLM yet its difficulty in accurately identifying event hallucinations, *this underscores the intrinsic complexity of event hallucinations and Hal-Evaluator's reliability in detecting event-type hallucinations.*

**Impact of LVLM output length on Generative Hallucination.** We investigate the potential correlation between the output length of LVLMs and the occurrence of generative hallucinations. We conducted the following experiments on LLaVA 1.5 and InstructBLIP. Experiments were carried out on LLaVA 1.5 and InstructBLIP wherein both models were prompted to describe the content of images with outputs of varying token lengths, using the Hal-Evaluator to detect and analyze the outputs at each length. The proportions of different types of hallucinations and the accuracy of outputs without hallucinations are visualized in Figure 6. Combining the results in subsection 4.2.1, the following findings were observed:

> **Finding 2**: *The incidence of hallucinations, especially event hallucinations, increases with the length of the output. Length control becomes a crucial aspect of generative evaluations, affecting comparative performance trends among LVLMs under varied output lengths.*

## 4.3 Comparative Analysis of Discriminative and Generative Evaluations

As shown in Table 5, for event hallucination, most LVLMs perform better in discriminative evaluation compared to other hallucination types. For example, we calculated the average F1 scores for discriminative evaluations across all models for different types of hallucinations, and the average F1 score for event types was 54.3, which is higher than that for other hallucinations. This suggests that models are more resistant to event-type hallucinations. However, when evaluating the models using a generative evaluation approach, we observed a higher frequency of event-type hallucinations in longer output sequences, contradicting the results from discriminative evaluations. We believe this is mainly because event-type hallucinations often contain more complex and rich information inconsistent with the image content, making them easier for models to handle in discriminative evaluations. However, this does not accurately reflect the models' ability to avoid event-type hallucinations effectively. Herefore, generative evaluation is a more effective method for assessing event-type hallucinations. For other types of hallucinations, such as objects, attributes, and relationships, research [20][29][14] indicates that discriminative evaluations are sufficiently effective in reflecting whether models tend to such hallucinations. In summary, we have the following findings:

> **Finding 3**: *The suitability of evaluation methodology varies according to the type of hallucinations. Using both discriminative and generative evaluations together gives a fuller view of tendencies to LVLM hallucination.*

## 4.4 Utilizing Hal-Data for Supervised Fine-tuning

To validate whether Hal-Data can assist LVLM in eliminating hallucinations and improving performance through instructional fine-tuning, we conducted the following experiment: we constructed instructional data from Hal-Data 130K, and after combining this instructional data with that of LLaVA1.5, we conducted joint fine-tuning training and get the varient of LLaVA1.5 named **Hal-VL**.

**Hallucination Benchmark:** As shown in Table 6 , we evaluate

| Method | Overall Score ↑ | Hallucination Rate ↓ | Score in Each Question Type ↑ | | | | | | | |
|---|---|---|---|---|---|---|---|---|---|---|
| | | | Attribute | Adversarial | Comparison | Counting | Relation | Environment | Holistic | Other |
| LLaVA-RLHF[7B] [39] | 2.05 | 0.68 | 2.92 | 1.83 | **2.42** | 1.92 | 2.25 | 2.25 | 1.75 | 1.08 |
| LLaVA[7B] [27] | 1.55 | 0.76 | 1.33 | 0.00 | 1.83 | 1.17 | 2.00 | 2.58 | 1.67 | 1.83 |
| Hal-VL | 2.12 (↑ 0.56) | 0.60 (↓ 0.16) | 2.84 | 2.11 | 2.17 | 1.74 | 2.05 | 2.44 | 1.66 | 1.62 |

**Table 7: Evaluation results for different MLLMs on MMHal-Bench.**

| Method | #Params | General VQA | | General VQA (Zero-shot) | | |
|---|---|---|---|---|---|---|
| | | VQAv2 | GQA | VizWizQA | TextVQA | SciQA |
| InstructBLIP [9] | 8.2B | - | 49.2 | 34.5 | 50.1† | 60.5 |
| Shikra | 7.2B | 77.4 | - | - | - | - |
| Qwen-VL-Chat [3] | 9.6B | 78.2 | 57.5 | 38.9 | 61.5‡ | **68.2** |
| LLaVA [26] | 7.2B | 71.3 | 41.3 | 36.7 | 50.2† | 61.5 |
| MiniGPT-4 [47] | 7.2B | 65.2 | 30.8 | 30.2 | 52.3† | 58.4 |
| LLaVA1.5 [24] | 7.2B | 78.5 | 62.0 | 50.0 | 58.2† | 66.8 |
| Hal-VL | 7.2B | **79.3** | **62.8** | **50.7** | 60.1† | 68.1 |

**Table 8: Performance comparison on visual question answering. For VQA, accuracy is reported. Note that specialists are fine-tuned on each individual dataset. † denotes OCR inputs are utilized. ‡ indicates the model has trained on the dataset.**

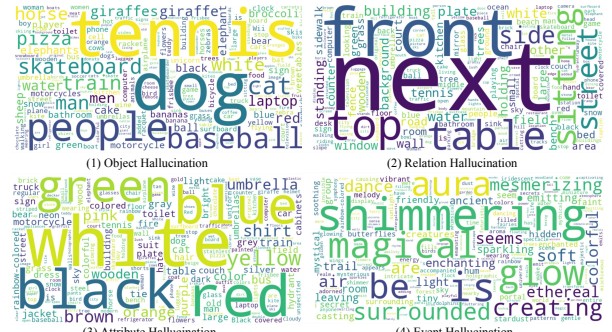

**Figure 7: We tokenized the annotations of hallucination content in Hal-Data 130K, and visualized them via a word cloud.**

Hal-VL on Hal-Eval and achieve the best performance. In addition to evaluations on Hal-Eval, as shown in Table 7, we also evaluate Hal-VL with other hallucination benchmark like MMHal-Bench [37]. The experiments demonstrate that Hal-VL significantly outperforms LLaVA 1.5. These results indicate Hal-Data effectively aid models in mitigating hallucinations.

**General benchmarks:** We also evaluate Hal-VL on multiple general benchmark such as VQA [11], GQA [16] and so on. As indicated in the Table 8, we found that Hal-VL achieved significant advantages on the vast majority of these benchmarks which suggesting Hal-Data can not only mitigating hallucinations but also enhances the overall model performance. These results support that:

> *Finding 4: The hallucinatory samples used to train our evaluator also serve as effective supervised fine-tuning data for LVLMs, contributing to the reduction of hallucinations and enhancement of their benchmark performance.*

## 4.5 Visualization for Hal-Data

As shown in Figure 7, we tokenized the annotations of hallucination content in Hal-Data 130K, and visualized them via a word cloud. The results shown in the word cloud align with the definitions of the hallucination content. For instance, the words in the word cloud for object hallucinations are distinguishable nouns, and for attribute hallucinations, they largely consist of features such as colors. This supports the reliability and validity of our labeled dataset.

## 5 RELATED WORK

**Large Vision Language Model:** Based on LLMs, there are three principal approaches to constructing LVLMs, all demonstrating potential for robust zero-shot generalization in the vision-language field. For instance, Flamingo [1] utilizes a fixed vision encoder paired with a sizable language model featuring gated cross-attention mechanisms for cross-modality matching. Meanwhile, PaLM-E [10] incorporates visual features via linear layers directly into the pre-trained PaLM [8] framework, which delivers strong performance across a spectrum of practical applications. This integration strategy is widely employed by models like LLaVA [27], Shikra [6], and others. However, generating long visual sequences remains a notable constraint of this technique. To mitigate this, BLIP-2 [19], inspired by DETR [4], conceived a Q-former that effectively condenses the length of visual feature sequences. This concept has since been reflected in Kosmos-1 [15], mPLUG-Owl [45], and MiniGPT-4 [48]. **Hallucination Evaluation:** LVLM hallucination benchmarks specifically aim at non-hallucinatory generation or hallucination discernment. These benchmarks are classified according to the type of evaluation approach they follow: Discriminative (Dis) or Generative (Gen). **Discriminative Benchmark:** POPE [20], NOPE [29], and CIEM [14] are examples of discriminative benchmarks. Each of these benchmarks exclusively directs attention towards object hallucinations and utilizes accuracy as their primary evaluation metric. The metric is calculated by querying the presence of objects within images and comparing the model's responses with the ground truth. **Generative Benchmarks:** modern research predominantly accentuates generative benchmarks over discriminative ones. Whilst discriminative benchmarks focus mainly on object-level hallucinations, generative benchmarks widen their scope to encompass a more extensive range of hallucinations, such as attribute and relation hallucinations [13, 17, 23, 38, 44]. AMBER [43] emerges as a holistic benchmark that concludes both generative and discriminative tasks.

## 6 CONCLUSION

We introduce a new category, Event Hallucination, into the study of LVLMs and develop a unique evaluation framework leveraging advanced LLMs for data analysis. This approach enhances the understanding and mitigation of hallucinations in LVLMs, marking a significant step forward in assessing and improving model performance.

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
