# OpenReview forum: "Hal-Eval: A Universal and Fine-grained Hallucination Evaluation Framework for Large Vision Language Models"
_acmmm.org/ACMMM/2024/Conference — MM2024 Oral_

### Official Review · Reviewer_v2Go · 2024-05-23

**Rating:** 4
**Confidence:** 4

**Summary:**

The paper introduces a novel type of hallucination, termed event hallucination. It argues that prior multimodal hallucination detection methods are capable of recognizing object, attribute, and relation hallucinations but fall short in effectively detecting event hallucinations. To address this gap, an event hallucination dataset was constructed, and several contemporary large models were evaluated and fine-tuned for hallucination detection on this dataset. Furthermore, the study proposes two detection methodologies: generative and discriminative. The discriminative method can uniformly detect various types of hallucinations, while the generative method is also capable of providing the correct output.

**Strengths:**

A novel type of hallucination—event hallucination—is introduced, with a precise definition provided. The existence of this hallucination type in multimodal language models is also validated.
Two hallucination detection methods are proposed: discriminative and generative. The discriminative method employs a unified prompt template, enabling the detection of various types of hallucinations. The generative method involves instruction fine-tuning of LLaVA 1.5 on the proposed Hal-Data, effectively detecting different hallucination types and content while providing correct responses.

**Limitations:**

● In Section 2, there is an error in the figure citation; Figure 8 does not exist.
● The increased likelihood of hallucinations during out-of-domain testing can be attributed to the inclusion of the COCO dataset during the instruction fine-tuning phase. However, in Section 3.2.1, the description of the out-of-domain dataset does not mention the COCO dataset.
● mPLUG-owl and MiniGPT-4 are relatively older models. For verifying the presence of hallucinations, it would be more appropriate to use the updated models, mPLUG-owl2 and MiniGPT-5.
● The relationships among object, attribute, relation, and event hallucinations have not been validated. For example, in the case of "small stream," where "stream" is identified as an object hallucination and "small" as an attribute hallucination, it is conceivable that "small stream" might also represent an event hallucination, despite being categorized as an object hallucination in Figure 3. Moreover, since event hallucinations encompass object, attribute, and relation hallucinations, it is plausible that the occurrence of any one of these hallucinations could trigger an event hallucination.
● Why the proportion of event hallucinations increases with output length, whereas the proportions of other types of hallucinations decrease In Figure 6.

**Suitability:**

3

---

### Official Review · Reviewer_BdQB · 2024-05-23

**Rating:** 5
**Confidence:** 3

**Summary:**

The paper introduces a new type of hallucination in Large Vision-Language Models (LVLMs) called Event Hallucination. In particular, the authors propose the Automatic Fine-grained Hallucination Annotation (AFHA) pipeline using GPT-4, constructing the Hal-Data as training data in their Hal-Eval benchmark. These collected hallucinatory samples could be used to fine-tune LVLMs, contributing to reducing hallucinations and better results on discriminative and generative hallucination evaluation methods.

**Strengths:**

* The concept of Event Hallucination involves the intricate integration of vision and language, which is crucial when generating longer text outputs.
* The idea of Automatic Fine-grained Hallucination Annotation (AFHA) pipeline is interesting.
* The authors have conducted comprehensive analysis of the need for Event Hallucination, particularly focusing on the length of the output.

**Limitations:**

* In line 448, the authors mention that they fine-tune LLaVA 1.5 13B with their Hal-Data. However, the performance in Table 7 compare to the approaches using LLaVA 7B with other instruction tuning dataset? How can it be a fair comparison? The authors should carefully describe the model and the instruction data difference compare to the approaches in other hallucination benchmark.
* In Table 7, is there any reason why Hal-VL gets worse performance on Attribute, Comparison, Counting, Relation, and Holistic question type in MMHal-Bench compare to LLaVA-RLHF 7B?
* In Table 8, what does the symbol ‡ mean? It indicates that the model has been trained on the General VQA dataset, but why is it still under the zero-shot setting?
* The experiments in Table 7 (other benchmark) and Table 8 (other task) could make better discussions.

**Suitability:**

3

---

### Official Review · Reviewer_dj2u · 2024-05-24

**Rating:** 5
**Confidence:** 3

**Summary:**

The paper introduces a new type of hallucination in Large Vision-Language Models (LVLMs): Event Hallucinations. To better evaluate Event Hallucinations in LVLMs, the authors propose a novel and universal fine-grained hallucination evaluation framework that unifies discriminative and    generative evaluations. The framework details the applicability of both evaluation methods. The authors developed an automatic annotation pipeline to generate fine-grained hallucination data, addressing the gap in large-scale datasets. Leveraging LLaVA 1.5 and the new data, they fine-tuned a high-performance model, Hal-Evaluator, for generative evaluation. Additionally, the authors fine tuned  the Hal-VL model using the new large-scale hallucination data to better mitigate hallucinations and improve model performance.

**Strengths:**

1. Category Innovation: The paper pioneeringly introduces a new type of hallucination: Event Hallucinations. The experiments demonstrate that Event Hallucinations constitute a significant proportion of all known hallucination types. In Experiment 4.2.2, it was found that Event Hallucinations become more severe as the model's generated output length increases. Additionally, Experiment 4.1.2 shows that COT can reducediscriminative hallucinations in LVLMs, improving the effectiveness in handling relational and event hallucinations.
2. Data Innovation: The paper establishes a comprehensive automatic fine-grained hallucination  annotation process, enriching the data scale for both discriminative and generative evaluations. This fills a gap in the study of LVLM hallucinations. Experiment 4.4 proves that these new large- scale hallucination annotations can be used to fine-tune LVLM models to reduce hallucinations.
3. Evaluation Framework: The Hal-Eval evaluation framework includes both discriminative and generative evaluation results. For discriminative evaluation, the authors created new data using   AFHA and replaced the questions with more natural ones. For generative evaluation, the authors fine-tuned the Hal-Evaluator using the generated large-scale data, which provides more accurate and detailed assessments of LVLM-generated hallucinations, pinpointing the exact locations of     hallucinations in the text.

**Limitations:**

1. Data Missing: In the out  of  domain_eval_5k and in  of  domain_eval_5k datasets, the new    hallucination descriptions and hallucination type annotations are evident. However, in the Hal- Data 130K SFT dataset, only human inputs and GPT's simple responses about whether the    captions accurately describe the images are visible, without the newly annotated hallucination types and descriptions.

2. Insufficient Experiment Details: The experimental process in Experiment 4.1's discriminative evaluation is not detailed enough. The evaluation metrics are too many and complex, making them difficult to understand, and the authors did not provide a good explanation of these metrics. The experimental process is almost not introduced, leaving questions about how the evaluation   was conducted, whether the descriptions with this type of hallucination also included correct descriptions, and the proportions of these descriptions. These unclear experimental processes make the results confusing.

3. Lack of Confidence in Experimental Data: In Experiment 4.2.2, the authors directly presented the correlation between different benchmark results and human evaluation results without specifying the human evaluation results and the results of LVLM tested based on these benchmarks. The authors did not provide the specific calculation process, raising doubts about the data's authenticity and potentially reducing the experiment's persuasiveness.

**Suitability:**

3

---

### Official Review · Reviewer_Hhwc · 2024-05-28

**Rating:** 4
**Confidence:** 3

**Summary:**

The paper introduces a type of hallucination, event hallucination, and proposes a novel approach to addressing hallucinations in LVLMs, providing a more reliable toolkit for assessing and addressing the hallucinations of LVLMs.

**Strengths:**

1. The introduction of event hallucinations and the evaluation benchmark addresses a previously overlooked aspect of LVLM hallucinations.
2. The paper provides a detailed and systematic approach to generating and filtering hallucination data, constructing datasets, and evaluating LVLMs, including both discriminative and generative LVLMs.
3. The paper conducts extensive experiments with leading LVLMs, providing a thorough analysis of their performance in handling different types of hallucinations.

**Limitations:**

1. In Figure 2, more LVLMs can be tested to illustrate the prevalence of event hallucinations and their characteristic of increasing with token lengths.
2. In Table 2, some models, such as mPLUG-Owl, MiniGPT-4, and InstructBLIP, have similar performance on both in-domain and out-of-domain benchmarks. Out-of-domain data is sampled from web-based datasets like CC, SBU, and LAION. Many LVLMs have already been trained on these datasets, and it is better to use completely unseen data as an out-of-domain benchmark.
3. In Table 4, the three metrics of Hal-Eval-Gen are all less than 0.5, which may suggest that the method does not strongly align with human judgment.
4. What LVLM is used in Figure 1?
5. Some typos:
  1. Figure 4, “Extrinsic” should be “Event”?
  2. Figure 6 caption, “LLaMA1.5” should be “LLaVA1.5”?

**Suitability:**

2

---

### Meta-Review · Area_Chair_hgcR · 2024-07-03

**Recommendation:** Accept (Oral)
**Confidence:** 5

**Metareview:**

The paper introduces a type of hallucination, event hallucination, and proposes a novel approach to addressing hallucinations in LVLMs. This paper addresses a highly practical issue, and the authors have devised a comprehensive solution, achieving excellent performance. All reviewers have given positive scores. I believe this paper will provide significant insights to the field. Therefore, I recommend this paper for an oral presentation at MM.